# CO_2_ Abatement in the Steel Industry through Carbon Recycle and Electrification by Means of Advanced Polymer Membranes

**DOI:** 10.3390/membranes11110856

**Published:** 2021-11-04

**Authors:** Marija Sarić, Jan Wilco Dijkstra, Yvonne C. van Delft

**Affiliations:** The Netherlands Organisation for Applied Scientific Research TNO, Unit Energy Transition, Westerduinweg 3, 1755 LE Petten, The Netherlands; jan_wilco.dijkstra@tno.nl (J.W.D.); yvonne.vandelft@tno.nl (Y.C.v.D.)

**Keywords:** process study, steel production, CO_2_ abatement, membrane application

## Abstract

The potential of advanced polymer or hybrid polymer membranes to reduce CO_2_ emissions in steel production was evaluated. For this, a conceptual process design and assessment was performed for a process that is a combination of carbon recycling and electrification of the steel making process. The results indicate a CO_2_ avoidance of 9%. CO_2_ emissions were reduced by factor 1.78 when using renewable electricity according to the proposed scheme compared to feeding this renewable electricity to the electrical grid. The CO_2_ abatement potential of the studied concept is highly dependent on the CO_2_ conversion in the plasma torch. If CO_2_ conversion in the plasma torch could be increased from 84.4% to 95.0%, the overall CO_2_ avoidance could be further increased to 16.5%, which is comparable to the values reported for the top gas recycling blast furnace. In this case, the CO_2_ emissions reduction achieved when using renewable electricity in the proposed scheme compared to using the same electricity in the electrical grid increases a factor from 1.78 to 3.27.

## 1. Introduction

Recent international agreements on CO_2_ emission reductions facilitate the industry sector to move towards sustainable and environmentally friendly solutions. Steel production with emissions of approximately 2.09 tCO_2_/t _hot metal_ (where t indicates metric tons) contributes significantly to the greenhouse gas (GHG) emissions of the industry sector [1]. There is already a significant effort in the steel sector to decrease their emissions, which is realized through different programs such as ULCOS, AISI, POSCO, COURSE50 and so forth [2]. One of the concepts studied extensively is the possibility of carbon reuse in the blast furnace by blast furnace gas (BFG) recycling in a concept called top gas recycling blast furnace (TGR-BF). Reported savings in carbon consumption are 15–25% [3], depending on the scenario considered, with a CO_2_ emissions reduction of 12–15% CO_2_/t _hot metal_ [4]. However, in TGR-BF configuration, the blast furnace needs to be modified to allow the combustion of coal in the presence of pure oxygen instead of air or oxygen enriched air. In this way, BFG is rich in CO_2_ with low inert gas (N_2_) content. This enables easy separation of CO_2_ from BFG, and recycling of the CO rich stream to the blast furnace. 

In a recent paper [5], the potential of carbon reuse in the steel industry using polyPOSS-imide membranes for H_2_ separation was evaluated. This concept combines hydrogen production from the coke oven gas (COG) by means of membranes, and electrification facilitated carbon recycling by means of syngas production in a plasma torch (PT). The reported CO_2_ emissions’ reduction potential is 14%, which is comparable to the TGR-BF. It is important to note that, for this concept, an expensive modification of the blast furnace to enable oxygen feeding was not required. 

In the literature, several authors have evaluated the application of membranes as an alternative to amine-based capture for applications in the steel industry [6,7]. Here, CO_2_ membranes are evaluated primarily for the purpose of CO_2_ storage applications. Other authors have evaluated plasma torch technology in the steel industry, aimed at emission reduction through fuel switch and electrification [8]. This paper combines these two options in a novel scheme where the CO_2_ separated is utilized for dry forming in the plasma torch rather than being exported for underground storage. 

The concept study (Figure 1) evaluates the potential for CO_2_ emissions’ reduction using CO_2_ selective membranes for CO_2_ separation from BFG and electrification through dry reforming in a plasma torch. In this concept, after separation from BFG by means of a membrane system, CO_2_ is mixed with COG and sent to the plasma torch. The retentate stream, which is rich in CO/N_2,_ is used to meet the heat demand of the steel plant. Syngas produced in the plasma torch from the dry reforming of methane in the COG with CO_2_ that is captured with the membranes, is recycled to the blast furnace resulting in the reduction of Pulverized Coal Injection (PCI). This ultimately results in reduced CO_2_ emissions for the steel making process. 

The present work evaluates the utilization of membrane technology based on advanced polymer or hybrid polymer with metal-organic frameworks (MOF) to facilitate carbon reuse in the steel industry by CO_2_ separation. Typically, plasticization is CO_2_-induced, thus CO_2_ swells the polymer chains at high CO_2_ partial pressures, resulting in an increase in inter-chain spacing and chain mobility. The membrane in turn loses its intrinsic selectivity. The use of hybrid polymer MOF membranes could possibly improve the separation performance, overcoming the plasticization problems of the pure polymer systems in carbon capture [9,10,11]. 

## 2. Materials and Methods

Advanced polymer or hybrid polymer MOF membranes can be used for CO_2_ separation in various CCUS pre-or post-combustion schemes [11]. In this work, the technical feasibility of CO_2_ selective membranes integration in the steel industry was assessed by a conceptual process of design and modelling. The proposed concept combines carbon utilization and the electrification of the steel making process. For the assessment, three cases were defined: (i) a baseline case without emissions reduction; (ii) a reference case using “conventional” MEA absorption separation technology; and (iii) a case with implemented advanced polymer or hybrid polymer MOF membranes. Conventional MEA absorption was selected as the reference technology for comparison, since this has the same functionality and also does not require modification of the blast furnace as is the case with most alternative technologies. The simplified flow schemes of the reference process and the membrane-based process are depicted in Figure 2, where the emissions reduction measures added to the baseline process are indicated in green. Residual gas from BFG is sent to a separation section (with or without membrane) where gas separation is performed. The produced CO_2_ rich stream is mixed with COG and sent to the plasma torch. In the plasma torch, renewable electricity is used to produce syngas for the blast furnace by dry reforming. This integration reduces the amount of PCI required in the blast furnace. Following this approach, the COG is not used anymore to cover the heat demand of the steel plant. That requires utilization of both CO/H_2_/N_2_ available from the BFG after the CO_2_ separation, and natural gas import to supply heat. In addition, the amount of BFG that is sent to the electricity production is reduced, resulting in a lower electricity production in the power plant. Thus, in addition to the renewable electricity used for the plasma torch, the import of renewable electricity is required to cover the electricity demand of the steel making process. Therefore, the proposed concept is a combination of a scheme for CO_2_ emission reduction and carbon reuse, and partial electrification of the steel making process. The reference case process is identical to the case with membrane technology with the difference being that CO_2_ is separated by a monoethanolamine (MEA) absorption based system.

The baseline case is conventional steel production in an integrated steel mill without CO_2_ capture, for which information on the feed/product streams was taken from literature [12]. The key data are coal and iron ore inflow, steel production capacity, CO_2_ emissions from steel and electricity production, total electricity produced, electricity and heat demand of the steel mill and residual steel gases flow rates and composition. Table 1 lists the key data for the baseline case. In the literature case, it is assumed that the plant is energy neutral, meaning that the gasses produced are just sufficient to cover the heat and power demand of the plant. 

The composition, flow and operating conditions of the BFG and COG are listed in Table 2. For the technical evaluation, mass and heat balances were made using the flow sheeting tool *Aspen Plus V10* with the RKSMHV2 thermodynamic method. 

From the simplified flowsheet given in Figure 2, a more detailed flowsheet is constructed (Figure 3). For the reference case, the BFG is sent to the CO_2_ capture unit, in which CO_2_ is absorbed using an conventional MEA absorber/regeneration unit. Produced lean BFG is used for internal heat generation. The rich amine stream is sent to a desorption unit for regeneration. The regeneration unit uses low-temperature steam. In this study, two scenarios for the reference case were evaluated: (i) Scenario 1: waste heat for steam generation is available at the steel plant; (ii) Scenario 2: waste heat is not available at the steel plant. The separated CO_2_ is mixed with COG and fed to a plasma torch for dry reforming. The plasma torch uses imported renewable electricity. The produced syngas is sent to the blast furnace, with the final aim of reducing PCI. 

When membranes are used for the CO_2_ separation, the BFG needs to be compressed, cooled and fed to the membrane separation unit. The units are operated with vacuum at the permeate side of the membrane, so the permeate CO_2_ rich stream needs to be re-compressed before mixing with COG. In addition, for the membrane case it is important to limit N_2_ crossover to the permeate side. Increased N_2_ crossover results in the accumulation of the inert gases as a result of the blast furnace gas recycle loop. This would result in increased compression power and an increase of the membrane surface area. Therefore, the two scenarios presented in Figure 4, when applying one or two stage membrane configurations, were evaluated, with the main goal of enhancing CO_2_ recovery and minimizing N_2_ crossover to the permeate side. 

Standard models were used for the heating, cooling and compression required. Assumptions used in the standard model are summarized in Table 3. 

### 2.1. Membrane Model

The membrane model used was a 1-dimensional cross-flow model using the approach of [15], which is implemented as an *Aspen Custom Modeller* sub-module in *Aspen Plus*. A schematic overview of the model is given in Figure 5. 

Here, the trans-membrane flux for each component i is described as a function of the dimensionless membrane length z according to Equation (1).
(1)$$J_{i}\left(z,T\right)={\;Q}_{i}\;\left[p_{i,f}\left(z\right)-p_{i,p}\left(z\right)\right],$$
with Q_i_ the permeance for component i and *p_i,f_(z)* and *p_i,p_(z)* the local partial pressure of component *i* at the feed and permeate side, respectively. An overview of the permeance data for the various gaseous components used in this study is presented in Table 4. Permeances assumed for the study are considered representative for polyactive and MOF-polyactive membranes. The permeance values assumed are listed in Table 4 and are deducted from combining unpublished experimental results with a 150 nm thick hybrid polyactive membrane with 20%w MOF-74, and literature data on PolyActive membrane 1500PEO77PBT23 presented in References [16,17], for which the performance was found to be very similar.

The sweep flow was set to zero. The membrane operating temperature was fixed at 30 °C. An optimization of the feed pressure was performed up to the maximum experimentally evaluated pressure of 5 bara as the upper feed pressure limit [16,17]. 

### 2.2. MEA Absorption/Regeneneration Section Model

The MEA absorber/regenerator unit was modelled as a black box separator model with a heat demand. Estimation of the MEA solvent flow was conducted according to guidelines from reference [18], assuming a net solution loading of 0.43 mol/mol of amine (based on the CO_2_ partial pressure in BFG). The typical energy use for MEA regeneration is reported in the range of 3–7 MJ/kg CO_2_ captured depending on the CO_2_ concentration in the stream and the CO_2_ recovery that needs to be achieved. Here, a steam consumption of 0.12 kg steam/kg of MEA solution corresponding with 3.6 MJ/kg CO_2_ captured was assumed [18]. The CO_2_ recovery was fixed at 90%.

### 2.3. Plasma Torch Model

For the plasma torch reactor experimental data of a single-anode hydrogen thermal plasma jet from reference [19] was considered. This type of plasma torch with both electrodes in the torch body is commonly studied in waste processing and has been applied for gas heating in steel production. Since no standard model is available for a plasma torch, a construct model was developed for the plasma torch based on a combination of standard Aspen Plus unit operations using experimental data from [19]. First, CO_2_ that is captured by the membranes or the MEA absorber is mixed with COG. The amount of CO_2_ is adjusted to the required molar ratio of CO_2_ over hydrocarbons:(2)$$\frac{{\mathrm{CO}}_{2}}{{\mathrm{CH}}_{4}+2\;C_{2}H_{4}}=2/3.$$

The resulting CH_4_/CO_2_ rich mixture is preheated by heat exchange with the syngas product of the plasma torch. The plasma torch model consists of three reactors in series, which are used to tune the conversion to experimental values. Assuming that the plasma torch is overall adiabatic, the energy balance model block combines the energy effect of each of the reactions with sensible heat effects to calculate the required electricity input. All reactors operate at atmospheric pressure and at a temperature of 1150 °C. 

The reactors, respectively, have the following functionalities: The first reactor R1 simulates the initial equilibrium; temperature was adapted to match the experimental methane conversion;The second reactor R2 is used to adjust the CO_2_ outlet mole fraction;The third reactor R3 is used to adjust the carbon yield.

The reactors are implemented as Gibbs free energy minimization reactors with temperature restricted equilibria, where reactions are specified by excluding all components other than those participating. Specifications are listed in Table 5. Though the reactors are placed in series, the model equations for three reactors are solved in parallel. 

### 2.4. Key Performance Indicators

The CO_2_
*avoided* is defined as the reduction in CO_2_ emission per unit of hot metal produced (*e)* during CO_2_ capture (*e_clk_*) with respect to the baseline process (*e_clk,ref_*) according to:(3)$${\mathrm{CO}}_{2}\;avoided=\frac{e_{clk,ref}-e_{clk}}{e_{clk,ref}}\;\;\;[\%]$$

The *Green electricity CO_*2*_ reduction potential (GECRP)* is a measure of how effectively the renewable electricity is contributing to emissions reduction. It indicates how renewable electricity reduces CO_2_ emissions when applied to a specific industrial process. This is compared to the emission reduction when feeding the same amount of power into the grid, thereby reducing emissions caused by reducing the fossil fuel power generation that is part of the grid mix. In the study, continuous renewable electricity supply is assumed to be enabled by deploying intermittent electricity sources and energy storage. The *GECRP* is defined as the amount of CO_2_ avoided per amount of renewable electricity used in the concept according to:(4)$$GECRP=\frac{{\dot{m}}_{{\mathrm{CO}}_{2},ref}-{\dot{m}}_{{\mathrm{CO}}_{2}}}{\Delta E_{import}}\;\;\;$$
where *ṁ*CO_2_, ref and *ṁ*CO_2_, ref are the mass flows CO_2_ emitted for the baseline and studied case respectively and Δ*E_import_* is the difference between imported power in the studied cases and the baseline case. The GECRP is expressed in kgCO_2_/GJ_el_ and is compared to feeding the renewable electricity in the electricity grid using typical CO_2_ emissions of 73 kg CO_2_/GJ_e_ from the average EU28 electricity mix of 2014 [20].

## 3. Results and Discussion

One and two-stage configurations were evaluated in detail for the integration of CO_2_-selective membranes. The amount of CO_2_ that needs to be mixed with the COG as specified by Equation (2) was found to be significantly lower than the value present in the BFG that is typically sent to the electricity production. Therefore, part of the membrane feed is based from the membrane section where The amount of BFG that is by-passed depends on the overall CO_2_ recovery in the membrane system and the amount of CO_2_ required for dry reforming. A sensitivity study on the overall CO_2_ recovery from 0.3–0.85 and on the operating pressure in the range from 2–5 bar was performed. 

As was mentioned, N_2_ will dilute the CO_2_ content in BFG and will therefore have a negative impact on the performance of the membrane based concept such as higher energy and membrane surface area requirement. A one-stage membrane configuration is not capable to sufficiently limit the N_2_ content in the CO_2_ stream. Depending on the overall CO_2_ recovery, the N_2_ content in the separated CO_2_ stream after the one-stage membrane varies from 8.3 to 11 mol% dry basis, for overall CO_2_ recoveries between 0.41 and 0.75. For the two-stage membrane, it was found that the N_2_ content in the CO_2_ permeate stream can be significantly decreased to 0.62–0.79 mol% on a dry basis for the same recovery range. Therefore, a two-stage membrane configuration, as shown in Figure 4, was selected for further evaluation.

The results of the sensitivity studies on the impact of overall CO_2_ recovery on the required total membrane surface area and the total compression duty for the two-stage membrane configuration are presented in Figure 6, Figure 7 and Figure 8. Figure 6 depicts the total compression power of the membrane section and the total membrane surface area vs. total CO_2_ recovery at different feed and permeate pressures and CO_2_ recovery in the second membrane stage. The power requirement decreases with an increase in the overall CO_2_ recovery, as a result of a lower amount of BFG that is required to produce a stochiometric amount of CO_2_ for the plasma torch. This can be clearly observed in Figure 7 and Figure 8, which show that the feed compressor has the highest share in the power demand of the membrane system. This power demand decreases with an increase of the overall CO_2_ recovery. The total power requirement decreases by 50% when the feed pressure is decreased from 5 bar to 2 bar. However, in this case, the maximum overall CO_2_ recovery is limited to 0.56, while at 5 bar feed pressure an overall CO_2_ recovery of 0.75 can be achieved. As expected, the membrane surface area increases with an increase in the overall CO_2_ recovery. The total membrane surface area increases by a factor 3.7 when the feed pressure is reduced from 5 to 2 bar. 

The total membrane surface area decreases by 37% when the permeate pressure for either the 1st or the 2nd stage is decreased from 0.3 to 0.1 bar. The total required compression power then increases by 23%. Additionally, investment costs of the vacuum pump will increase significantly when decreasing the permeate pressure to 0.1 bar because of the large volumetric feed flow.

Increase of the CO_2_ recovery at the 2nd membrane stage results in a decrease of the required membrane surface area and compression power. However, at CO_2_ recoveries higher than 0.7, a minor impact on the further decrease of the membrane surface area and power consumption was observed. 

Two subcases, GC-100 and GC-101, were selected for the further evaluation of integration in the steel making plant. These two subcases differ in membrane feed side pressure for both the first and second stage which is: 5 bar for GC-100 and 2 bar for GC-101. From the results in Table 6, it can be seen that, in terms of the PCI reduction potential, these two subcases are comparable. For the 5 bar feed pressure operation 1.44 GJ/t is required to capture CO_2_. This value reduces by 30% for 2 bar feed pressure operation. However, the total membrane surface area for the low pressure operation of case GC-101 is a factor 3.7 higher. Further economic evaluation using these results is required in order to determine which of these subcases is optimal in economic terms. 

The results of the reference case using MEA absorption for separation of CO_2_ are presented in Table 7. The MEA absorption provides a higher CO_2_ recovery and a higher CO_2_ purity. This resulting PCI reduction potential is comparable to that of the membrane case, with the slight difference coming from the lower slip of other components such as H_2_ and CO in the CO_2_ stream of the reference case. 

Table 8 lists the overall system performance results for the reference and membrane cases in comparison with the baseline no capture case. Since the CO_2_ conversion that can be achieved in the plasma torch is a uncertain parameter and was observed to have a negative impact on the PCI reduction potential, results for both the reference and membrane cases are given for two CO_2_ conversion values: (i) 84.37% that is experimentally reported in the literature (baseline PT conversion) and (ii) 95% that could be potentially achieved (high PT CO_2_ conversion) [19]. 

There is a significant impact of the overall CO_2_ conversion increase in the plasma torch on CO_2_ separation cases results. If it is possible to increase the CO_2_ conversion from 84.37% to 95%, the overall CO_2_ avoided can be increased from 753 kt/year to 1373 kt/year for the membrane cases. Respectively, the total carbon input (sum of coking coal and PCI inflow) to the steel production decreases by 14% or 26%. This is comparable to the results reported for TGR-BF [3], but without the need for the modification of the blast furnace.

The CO_2_ emission reduction potential is slightly lower for the reference case compared to the membrane cases. If waste heat is available for the solvent regeneration (Reference Scenario 1), the reference case has the advantage of having a lower power demand than the membrane cases and then the total energy use is slightly lower for the reference case. If this heat is not available, (Reference Scenario 2) the reference case has a lower energy reduction potential compared to membrane cases. Moreover, the CO_2_ avoided is lower since CO_2_ emissions are generated by natural gas consumption for the heat generation. 

The green electricity CO_2_ reduction potential (GECRP) for all studied cases is higher than for the EU28 mix. The GECRP is the highest for Reference Scenario 1, because the avoided CO_2_ is comparable to the membrane cases, but the green electricity import is lower because it profits from the waste heat available at the steel mill site. The utilization of renewable electricity according to the membrane cases or the reference case instead of feeding this renewable electricity to the grid as for the assumed mix, yields into CO_2_ emission reduction per GJ, which is 1.79 and 1.92 times higher than the mix, respectively. These numbers increase to 3.52 times for the reference case (Reference Scenario 1), and 3.27 times for membrane cases with 95% CO_2_ conversion in the plasma torch. In the case where heat for solvent regeneration is not available (Reference Scenario 2), a slightly lower CO_2_ avoidance potential is found (compared to the membrane cases studied) and its GECRP is comparable to the membrane cases. 

## 4. Conclusions

Advanced polymer or hybrid polymer MOF CO_2_ separating membranes are used in the steel industry using renewable power to ultimately reduce pulverized coal injection. For this blast furnace, gas is sent to a membrane system producing CO_2_ stream that, in combination with cokes oven gas, undergoes dry reforming in a plasma torch. The syngas stream produced is fed to the blast furnace resulting in a reduction of pulverized coal injection required for the steel production. MEA absorption was selected as the reference technology for CO_2_ removal. For both schemes, the mass and energy balances have been developed based on a baseline steel plant from the literature and unit operation models for all relevant equipment. In both cases, a significant amount of renewable power imported to the plant is used in the plasma torch, for the compression of gases and to compensate for a lower power production in the steel plant because less power is produced by the BFG. The membranes concept allows 9% CO_2_ emissions avoidance equal to 740–753 ktCO_2_/year avoided, depending on the process conditions selected. The reference case has a lower power demand and a higher heat demand, which together results in a comparable CO_2_ emission avoidance of 8.9%. The heat demand could possibly be supplied by waste heat available at the steel mill, which would favor the reference case. The total required import of renewable electricity for the reference process is considerable at 183 MW. Both the membrane and reference cases make effective use of renewable electricity for emission reduction. The utilization of renewable electricity, according to the membrane case or the reference case scenario instead of feeding it to the grid, yields a CO_2_ emission reduction per GJ of 1.79 and 1.92 times higher respectively. If it is possible to increase CO_2_ conversion in the plasma torch from its current value of 84.37% to 95%, the CO_2_ avoidance could be increased to 15.23%–16.52%, corresponding to a total CO_2_ emissions’ avoidance of 1265–1373 kt/year. The utilization of renewable electricity in this case results in 3.52 and 3.27 times lower CO_2_ emissions per GJ compared to feeding the same amount of renewable electricity to the grid. Modelling of the hybrid polymer MOF membrane technology integration into the steel production demonstrated significant CO_2_ emissions’ reduction potential. 

Further evaluation of the potential study and the difference between polyactive and polyactive MOF membranes requires more detailed experimental work and an assessment of membrane lifetime. Some of the recent studies show that a large environmental burden can be allocated to the MOF production [21,22]. In order to evaluate the full potential of this technology, further combined economic and environmental assessments need to be considered.

## Figures and Tables

**Figure 1 membranes-11-00856-f001:**
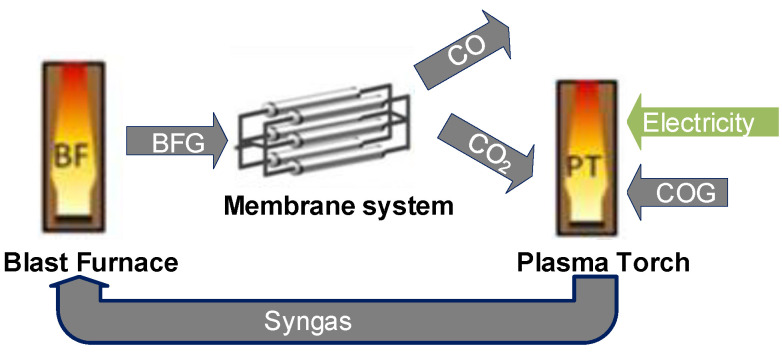
Concept considered in the study.

**Figure 2 membranes-11-00856-f002:**
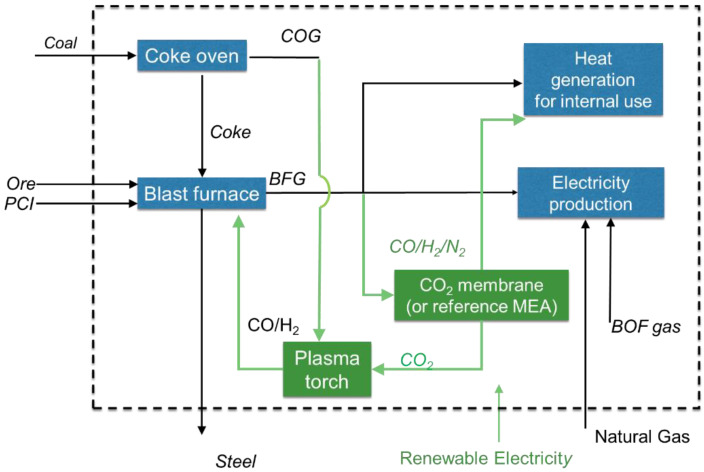
Simplified process schematic for the membrane and reference process.

**Figure 3 membranes-11-00856-f003:**
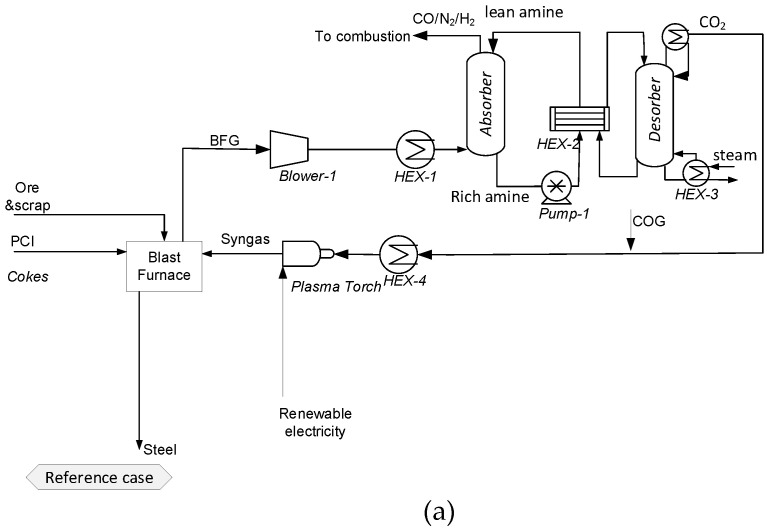
Process flow diagram of the (**a**) Reference case and (**b**) Membrane configuration.

**Figure 4 membranes-11-00856-f004:**
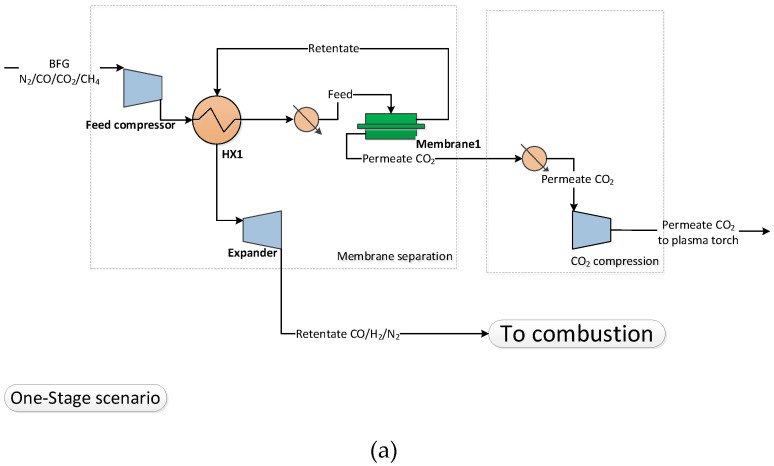
Process flow diagram of (**a**) one and (**b**) two stage separation scenarios.

**Figure 5 membranes-11-00856-f005:**
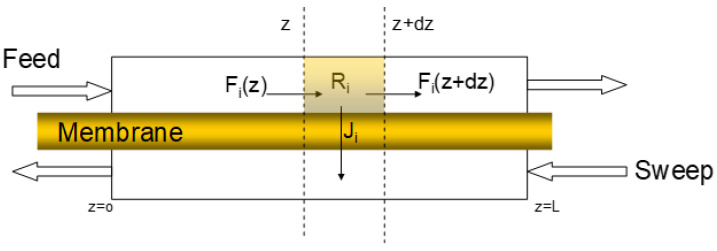
Schematic overview of the membrane model.

**Figure 6 membranes-11-00856-f006:**
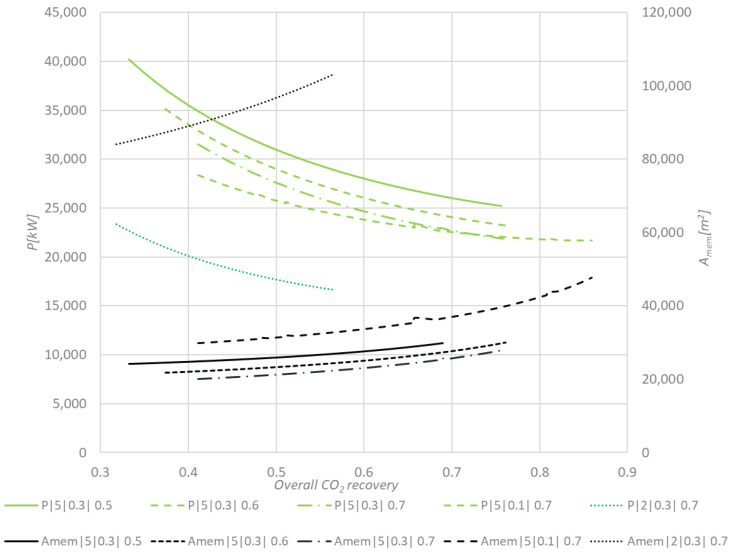
Sensitivity study: Compression power (P) and membrane surface area (A_mem_) as a function of the overall CO_2_ recovery. Legend format: p_feed_**|** p_permeate_| CO_2_ recovery in the second-membrane stage. CO_2_ recovery in the first stage is fixed at 0.7.

**Figure 7 membranes-11-00856-f007:**
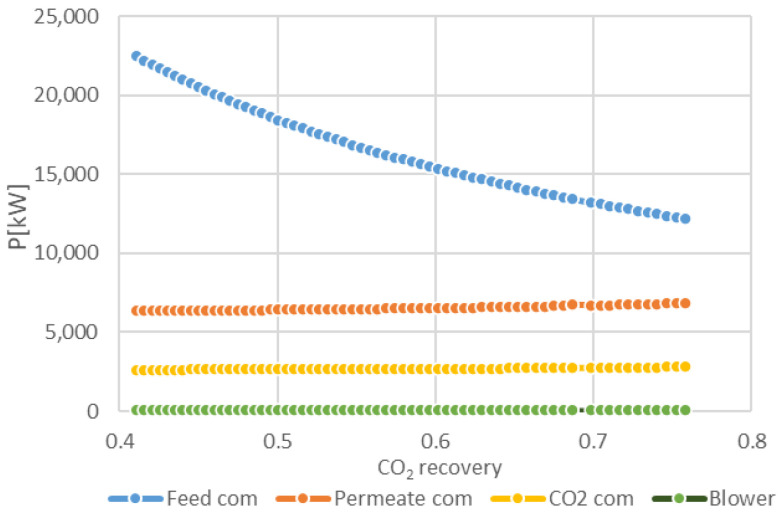
Breakdown of compression power (P) at different overall CO_2_ recovery (at p_feed_ = 5 bara, p_perm_ = 0.3 bar, 2nd stage CO_2_ recovery 0.7).

**Figure 8 membranes-11-00856-f008:**
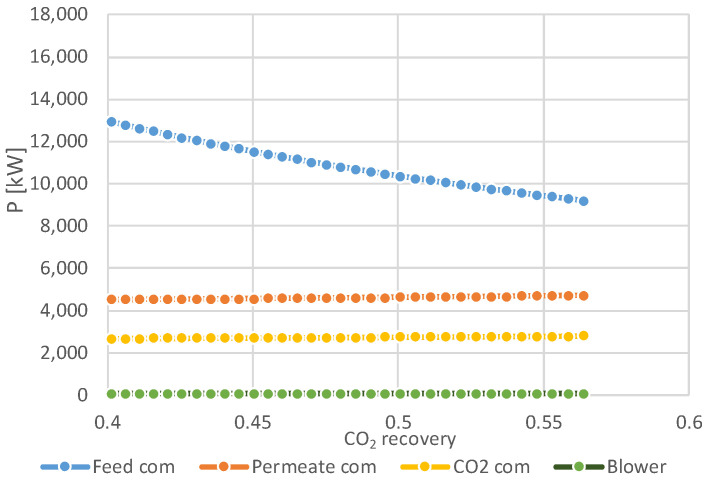
Breakdown of compression power as function of overall CO_2_ recovery (at p_feed_ = 2 bara, p_perm_ = 0.3 bar, 2nd stage CO_2_ recovery 0.7).

**Table 1 membranes-11-00856-t001:** Key data for the baseline steel mill [12].

Parameter	Value
Hot metal (HM) production capacity [Mt/year]	4.0
Use of coking coal [GJ/t HM]	16.292
Use of PCI [GJ/t HM]	5.032
Natural gas import for electricity production [GJ/t HM]	0.85
Overall CO_2_ emissions [kg/t of HM]	2094.4
Power plant average daily power output [MW_el_]	200
Export of power [MW]	0

**Table 2 membranes-11-00856-t002:** BFG and COG composition and flow.

Component/Conditions	Unit	BFG	COG
CO_2_	[mol%]	22.1	0.94
CO	[mol%]	22.34	3.84
N_2_	[mol%]	48.77	5.74
H_2_	[mol%]	3.63	59.54
CH_4_	[mol%]	-	23.04
O_2_	[mol%]	-	0.19
H_2_O	[mol%]	3.15	3.98
Other HC	[mol%]	-	2.69
P	[bara]	1.11	1.11
T	[°C]	25	29
Phase	V/L/S	V	V
Total flow	[kNm^3^/h]	791	80
Flow to power plant	[kNm^3^/h]	464	0

**Table 3 membranes-11-00856-t003:** Assumptions used for modelling standard equipment [13,14].

Parameter	Unit	Value
Temerature of cooling water	[°C]	18
Cooling water max. temperature increase	[°C]	10
*Assumed T differences in heat exchangers*		
Gas/gas	[°C]	25
Gas/boiling liquid or liquid	[°C]	10
Liquid/liquid	[°C]	10
Condensing /liquid	[°C]	3
*Assumed pressure drop in heat exchangers*		
Liquid phase for hot/cold side	[bar]	0.4
Gas phase	[%]	2
*Efficiency of compressors/expanders*		
Isentropic efficiency	[%]	85
Mechanical efficiency	[%]	95

**Table 4 membranes-11-00856-t004:** Membrane permeance data at T = 30 °C used in the system study.

Component	Qi	
	mol/m^2^sPa	GPU
CO_2_	1.02 × 10^−7^	693
H_2_	2.27 × 10^−8^	68
CO	8.98 × 10^−9^	27
N_2_	4.33 × 10^−9^	13

* GPU = gas permeation unit, 1 GPU = 10^−6^ cm^3^(STP)/(cm^2^ s cm Hg).

**Table 5 membranes-11-00856-t005:** Plasma torch model specifications.

Reactor	Target	Target Value	Temperature Approach on
R1	CH_4_ conversion	87.98%	CH_4_, CO_2_, CO, H_2_O, H_2_, C_2_H_4_
R2	CO_2_ conversion	84.37%	CO_2_ + H_2_ 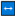 CO + H_2_O
R3	CO conversion	17.66%	2 CO 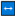 C + CO_2_

**Table 6 membranes-11-00856-t006:** Modelling results for two scenarios of CO_2_ separation utilizing membrane technology.

Case	GC-100	GC-101
Parameter	Value	
Membrane feed pressure, 1st & 2nd stage [bar]	5	2
Membrane permeate pressure, 1st & 2nd stage [bar]	0.3	0.3
Total CO_2_ removed [kt/year]	421	421
Overall CO_2_ recovery [%]	75	56
A_mem_ (Membrane 1) [m^2^]	24872	93051
A_mem_ (Membrane 2) [m^2^]	3004	9906
Feed compressor [kW]	12173	7900
Vacuum compressor (stage-1) [kW]	2806	2787
Compressor permeate (stage-1) [kW]	4036	1917
Vacuum compressor (stage-2) [kW]	1525	1418
Retentate recycle compressor [kW]	52	147
COG blower [kW]	425	425
Energy/CO_2_ captured [GJ/t CO_2_]	1.44	0.96
Plasma torch power demand [kW]	57625	57590
Plasma Torch PCI reduction potential [kg/t _hot metal_]	73	74

**Table 7 membranes-11-00856-t007:** Modelling results for the reference case.

Parameter	Value
CO_2_ recovery [%]	90
Total CO_2_ removed [kt/year]	421
Steam (@3.5 bar) [MW]	52.8 (3.6 GJ/tCO_2_ captured)
COG blower [kW]	425
PT [kW]	56699
Plasma Torch PCI reduction potential [kg/t _hot metal_]	72

**Table 8 membranes-11-00856-t008:** Comparative modelling results for the reference case and the case integrating the novel membrane technology.

Parameter	Unit	No Capture	Reference Scenario 1 *	GC-100	GC-101	Reference Scenario1 *	Reference Scenario 2 **	GC-100	GC-101
			Baseline PT CO_2_ conversion			High PT CO_2_ conversion			
HM production	Mt _hot metal_/year	3.97	3.97	3.97	3.97	3.97	3.97	3.97	3.97
CO_2_ conversion PT	[%]	-	84.37%	84.37%	84.37%	95%	95%	95%	95%
Coking coal	kg/t _hot metal_ (GJ/t _hot metal_)	354.8 (16.29)	354.8 (16.29)	354.8 (16.29)	354.8 (16.29)	354.8 (16.29)	354.8 (16.29)	354.8 (16.29)	354.8 (16.29)
PCI	kg/t _hot metal_ (GJ/t _hot metal_)	152	80 (2.64)	79 (2.61)	78 (2.60)	19 (0.64)	19 (0.64)	18 (0.61)	18 (0.60)
PT syngas to BF	kg/t _hot metal_	0	152	153	154	152	152	147	148
Natural gas for electricity production	GJ/t _hot metal_	0.85	0.85	0.85	0.85	0.85	0.85	0.85	0.85
NG for steam production	GJ/t _hot metal_	0	0	0	0	0	0.43	0	0
Electricity import	MW_el_ (GJ/t _hot metal_)	0	183 (1.33)	205 (1.48)	199 (1.45)	184 (1.33)	184 (1.33)	207 (1.50)	199 (1.45)
Energy IN	GJ/t _hot metal_	22.17	20.1	21.2	21.2	19.1	19.5	19.2	19.3
Energy use reduction	%		4.80%	4.18%	4.46%	13.81%	11.89%	13.17%	13.49%
CO_2_ emissions	kg/t _hot metal_	2094	1908	1903	1904	1751	1776	1750	1748
CO_2_ avoided	%	n.a.	8.9%	9.0%	9.1%	16.4%	15.2%	16.5%	16.5%
CO_2_ avoided	kt/year	0	740	750	753	1360	1265	1368	1373
GECRP	kgCO_2_/GJ_el_	n.a	140	126	131	257	239	229	239

* Waste heat available at the steel mill, ** Waste heat not available at the steel mill.

## Data Availability

Not applicable.

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
