# Peer review of "CO2 Abatement in the Steel Industry through Carbon Recycle and Electrification by Means of Advanced Polymer Membranes"

_membranes, 2021, doi:10.3390/membranes11110856_

Round 1
Reviewer 1 Report
In the work “CO2 abatement in the steel industry through carbon recycle and electrification by means of advanced polymer membranes”, the authors investigated the potential of advanced (hybrid) polymer membranes to avoid CO2 emissions in the steel production industry. The authors first implemented conceptual carbon capture and electrification sectors to the existing steel production process in Aspen Plus, and then studied the impacts of several design parameters on the system CO2 emission reduction performance. Compared to the baseline case (no CCS), both the MEA-CCS case and the polymer membrane-CCS case showed decent CO2 avoidance performance. Overall, the model development in Aspen seems solid and the obtained results are mostly well-explained. Under the global warming mitigation background, the work also has practical application value. Therefore, I would recommend the paper to be published after minor revision. Below are my comments that hope the authors could consider/addresses properly.
- The authors listed the membrane permeance data in Table 4. Are these data representing the classic PolyActive membranes or are they representing the unpublished results for a MOF-74/PolyActive membranes? If the permeance (and thus selectivity) data are similar for the two types of membrane, then how would the use of hybrid membrane improve the separation performance (line 61-63)? I hope the authors could add more clarifications in between line 159 and line 165.
- From the results listed in Table 8, it seems that MEA-based and membrane-based CCS are similar in performance. Given that MEA-based CCS is well-established, I was wondering if the authors could comment more on when the (hybrid) membrane-based CCS would become more promising than the MEA-based CCS?
- Please double-check the citation of Ref 23. If I found the right paper, it should be Sustainable materials and technologies, 2017, 14, 11-18. The year, volume and page number are missing in the current citation.
Author Response
Detailed response to reviewers' comments
- Manuscript Number: 1300870
- Title: CO2 abatement in the steel industry through carbon recycle and electrification by means of advanced polymer MOF membranes
We would like to thank the reviewers for their remarks and have revised the manuscript accordingly. Below you can find a short overview of the comments of the reviewers and the respected changes made to the article.
Detailed response to reviewers' comments
- Manuscript Number: 1300870
- Title: CO2 abatement in the steel industry through carbon recycle and electrification by means of advanced polymer MOF membranes
We would like to thank the reviewer for remarks and we have revised the manuscript accordingly.
For reply to the comments please see the attachment.

Reviewer 2 Report
- In this paper, the author pointed out that the the potential of advanced polymer or hybrid polymer membranes to reduce the CO2 emissions in the steel production was evaluated. In the manuscript there is not information about the properties of the membranes and the characterizations. I suggest the author explain further with a paragraph.
- In the introduction, it is suggested to make a summary of similar literature. in the summarized section, a comparison based on reference information is suggested. The summary and comparison would give the reader a whole picture and the trend in research.
- The novelty of the work is to be clarified more.
- It is necessary to clear Figure 1. This is a probably graphical abstract of the work.
- The reason and the mechanisms that cause the redaction of CO2 in the electrical grid are suggested to be clarified more.
- It is necessary to divided the section of material and method for clarify the experimental section.
Author Response
Detailed response to reviewers' comments
- Manuscript Number: 1300870
- Title: CO2 abatement in the steel industry through carbon recycle and electrification by means of advanced polymer MOF membranes
We would like to thank the reviewer for remarks and we have revised the manuscript accordingly. For response to the comments Please see the attachment.

Reviewer 3 Report
The main objective of this manuscript is developing an advanced membrane-based system to reduce the CO2 emissions in the steel production. A conceptual process design and assessment was performed involving a combination of carbon recycle and electrification of the steel making process. Authors found that this system is a highly efficient platform for CO2 abatement in the steel industry. After reading the full paper, there are still many points not covered in the paper, so the manuscript needs to be revised again corresponding to the comments below:
In the Materials and Methods section, the authors pointed out that “the case with implemented advanced polymer or hybrid polymer MOF membranes.” . As is well known, there are many kinds of advanced polymer or hybrid polymer MOF membranes. Here, the specific chemical composition and morphology of the membranes should be described exactly.
Author Response
- Manuscript Number: 1300870
- Title: CO2 abatement in the steel industry through carbon recycle and electrification by means of advanced polymer MOF membranes
We would like to thank the reviewer for remarks and we have revised the manuscript accordingly. Please see the attachment for the answers to the comments.

Round 2
Reviewer 2 Report
The manuscript is interesting and ready for the publication.
This manuscript is a resubmission of an earlier submission. The following is a list of the peer review reports and author responses from that submission.
Round 1
Reviewer 1 Report
In the work “CO2 abatement in the steel industry through carbon recycle and electrification by means of advanced polymer MOF membranes”, the authors conduced a conceptual design and assessment of a combination of carbon cycle and electrification of steel production in Aspen environment. Three cases, a baseline case, a MEA-CCS case, and a MOF-polymer-CCS case, were analyzed, and the influences of several design parameters on the system CO2 emission reduction performance were investigated and discussed. Overall, it seems a solid piece of work with practical application value. I would recommend the paper to be published after revision. Below are my comments that hope the authors could consider/addresses properly.
- In the process evaluations, renewable electricity was heavily imported to the system. Is this a good/practical assumption as of now, given the energy infrastructure is still dominated by fossil fuels in most regions of the world? Also, the steel production is regarded as a continuous process, yet the renewable energy production (and thus its supply to the power grid) follows an intermittent manner. Could the authors please comment on this point and justify the assumption on the renewable electricity?
- For the membrane part, the key information listed in Table 4 (obtained from REF 17 and 18) seem to be based on pure polymer membranes. Could the authors please include more details for the MOFs-polymer membranes (e.g., permeance, selectivity, MOFs type and MOFs loading) used in the Aspen simulation?
- It has been widely recognized that different types of MOFs have dramatically different CO2 permeance and CO2/N2 selectivity, which would affect the CO2 recovery ratio of the membrane separation module. Could the authors comment on how the results obtained from this work would vary with the type of MOFs? In other words, are the quantitative results representative enough?
- Some of the MOFs that have high CO2/N2 separation efficiency (e.g., HKUST-1) are sensitive to the water moisture in the separation target, and water vapor is presented in Table 5. Is the compatibility of the MOFs in the MOF membrane with the separation target considered? Is the MOFs in the membrane water stable? Such practical considerations are important for simulation studies.
- Some of the recent works highlighted the high environmental burdens with the MOFs production (Sustainable materials and technologies, 2017, 14, 11-18. Journal of Environmental Chemical Engineering, 2021, 9, 105159.), i.e., a large amount of CO2 is first generated before MOFs could be used for CO2 emission avoidance. When comparing to MEA, such environmental burdens associated with MOFs production would directly affect the “true” CCS effects of MOFs in process like NGCC-CCS (Journal of Environmental Chemical Engineering, 2021, 9, 105159.). In the present work, the authors suggested that larger membrane area is required under some operation conditions, and the use of large membranes inevitably leads to more MOFs being produced (and thus more CO2 being generated first). I strongly recommend the authors include those recent works and comment on the potential environmental limitation of using MOF polymer membranes (when compared to MEA), as this point is as important as economic considerations for future process design.
- Please make Figure 6 more readable.
Reviewer 2 Report
The manuscript is discussing a very interesting topic aiming at CO2 emission reduction from industrial sources. The manuscript is well-written and well-organized. Few comments and minor comments to consider are as follow:
1- Given the huge interest and ongoing efforts in CO2 emissions reductions in industrial sectors, can the author provide a comparison of the proposed solution to the recently developed solution (other than the well-established amine method used as a reference case).
2- Can you identify some of the abbreviations used such as HM, GPU...
Regards
Reviewer 3 Report
The manuscript „CO2 abatement in the steel industry through carbon recycle and electrification by means of advanced polymer MOF membranes” attracted my attention due to use of some keywords matching general trend in development of industry toward more environmentally friendly processes: membranes, MOF, carbon recycle.
Unfortunately, there is no “membrane” related information in the abstract and in the text of the manuscript, until the page 6, Table 4 where I had to stop reading.
Table 4 “Membrane permeance data based on experimental data for PolyActive membrane 1500PEO77PBT23 from [17, 18].” refers one to the membrane formed of block copolymer with selective layer designed for CO2/N2 separation. The PolyActive membrane contains no MOF ! To my surprise, values presented in the Table are not related to the reality! I’d like to ask authors to point out the source of the data since in references 17 and 18 there are no such values. The selectivity CO2/N2 is 1.25E06/3.19E07=3.92E-02, which is very interesting since for the PolyActive membrane at room temperature the value should be around 50. Nitrogen can't permeate by solution-diffusion mechanism faster than carbon dioxide! Everything is wrong in this table! What is GPU? I'm fighting to get CO2 permeance 2E03 GPU and dream of 3E03 GPU (please see for comparison page 9 of https://www.osti.gov/servlets/purl/1015458-INdTMC/ ) and here one finds value of 1.25E06!
Any use of values presented in the Table 4 would lead to massive errors in process modelling and based on this consideration I stopped reading the manuscript and would like to insist on manuscript withdrawal.